# Gamma Knife Radiosurgery Irradiation of Surgical Cavity of Brain Metastases: Factor Analysis and Gene Mutations

**DOI:** 10.3390/life13010236

**Published:** 2023-01-14

**Authors:** Yi-Han Huang, Huai-Che Yang, Chi-Lu Chiang, Hsiu-Mei Wu, Yung-Hung Luo, Yong-Sin Hu, Chung-Jung Lin, Wen-Yuh Chung, Cheng-Ying Shiau, Wan-Yuo Guo, Cheng-Chia Lee

**Affiliations:** 1Department of Neurosurgery, Neurological Institute, Taipei Veterans General Hospital, Taipei 112, Taiwan; 2School of Medicine, National Yang Ming Chiao Tung University, Taipei 112, Taiwan; 3Department of Chest Medicine, Taipei Veterans General Hospital, Taipei 112, Taiwan; 4Institute of Clinical Medicine, School of Medicine, National Yang Ming Chiao Tung University, Taipei 112, Taiwan; 5Department of Radiology, Taipei Veterans General Hospital, Taipei 112, Taiwan; 6Cancer Center, Taipei Veterans General Hospital, Taipei 112, Taiwan; 7Brain Research Center, National Yang Ming Chiao Tung University, Taipei 112, Taiwan

**Keywords:** brain metastasis, Gamma Knife, radiosurgery, surgical cavity, survival, tumor control

## Abstract

**Simple Summary:**

Stereotactic radiosurgery is widely used to improve tumor control in cases of brain metastases; however, there remains considerable disagreement as to whether radiation treatment following surgical resection provides any benefits in terms of tumor control or overall survival. Our objective in the current research was to elucidate the efficacy of post-operative stereotactic radiosurgery. We determined that administering stereotactic radiosurgery to surgical cavities improved tumor control; however, it did not appear to affect overall survival. We would encourage patients with brain metastasis to undergo stereotactic radiosurgery to the post-surgical cavity to improve tumor control.

**Abstract:**

(1) Background: Surgical resection for the removal of brain metastases often fails to prevent tumor recurrence within the surgical cavity; hence, researchers are divided as to the benefits of radiation treatment following surgical resection. This retrospective study assessed the effects of post-operative stereotactic radiosurgery (SRS) on local tumor control and overall survival. (2) Methods: This study examined the demographics, original tumor characteristics, and surgical outcomes of 97 patients who underwent Gamma Knife Radiosurgery (GKRS) treatment (103 brain metastases). Kaplan–Meier plots and Cox regression were used to correlate clinical features to tumor control and overall survival. (3) Results: The overall tumor control rate was 75.0% and overall 12-month survival was 89.6%. Tumor control rates in the radiation group versus the non-radiation group were as follows: 12 months (83.1% vs. 57.7%) and 24 months (66.1% vs. 50.5%). During the 2-year follow-up period after SRS, the intracranial response rate was higher in the post-craniotomy radiation group than in the non-radiation group (*p* = 0.027). Cox regression multivariate analysis determined that post-craniotomy irradiation of the surgical cavity is predictive of tumor control (*p* = 0.035). However, EGFR mutation was not predictive of overall survival or tumor control. (4) Conclusions: Irradiating the surgical cavity after surgery can enhance local tumor control; however, it does not have a significant effect on overall survival.

## 1. Introduction

Stereotactic radiosurgery (SRS) is an effective primary treatment for oligo-metastases (e.g., 1–10 lesions), providing good local control with minimal radiation toxicity. SRS can also be used as a preventive procedure, which involves the delivery of radiation to the post-surgical cavity to reduce the likelihood of marginal recurrence. For patients with a small number of large intracranial metastases, adjuvant SRS in the post-surgical cavity is generally well tolerated (e.g., quality of life) and does not interfere with the scheduling of systemic therapies.

Recent tyrosine kinase inhibitors (TKIs), such as gefitinib, erlotinib, afatinib, and osimertinib, have been shown to improve outcomes in cases of epidermal growth factor receptor (EGFR)-mutant non-small cell lung cancer brain metastasis (NSCLC-BM) [1,2,3]; however, an inability to penetrate the central nervous system is a limiting factor. This has promoted the use of SRS boost therapy in cases of progressive brain metastases. Previous studies showed that EGFR mutation status could have an effect on the response to radiotherapy, causing an impact on overall survival and tumor control. Our objective in the current study was to compare the effectiveness of surgery alone versus surgery plus adjuvant SRS to the post-surgical cavity in terms of local tumor control in cases of 1 or 2 brain metastases EGFR mutation status is also recorded and analyzed. We also examined the incidence of complications associated with preventive SRS irradiation of the surgical cavity.

## 2. Materials and Methods

### 2.1. Study Design

This retrospective review focused on consecutive patients who underwent SRS for brain metastasis at our institution between 2006 and 2022. Inclusion criteria included the following: (1) a confirmed diagnosis of one or two brain metastases based on open surgery on the original tumor; (2) treatment using SRS or whole brain radiotherapy (WBRT); and (3) at least one clinical and neuroimaging follow-up assessment. We reviewed the pre-SRS MR images carefully to assess tumor volume and characterize tumor components in Table 1. The Institutional Review Board waived patient consent due to the retrospective nature of the study and anonymizing of data.

A total of 97 patients satisfied all inclusion criteria (103 brain metastases). The study population included 49 males and 48 females with a median age of 59.2 years. Most of the patients (*n* = 91; 93.8%) presented single intracranial surgical cavity, and the others presented two or more surgical cavities. Thirty-nine patients (40.2%) had extracranial metastasis. The median Karnofsky performance status (KPS) was 90. The incidence of neurological symptoms was as follows: long tract signs (42.3%), cerebral signs (0%), cranial nerve palsy (10.3%), and high cortical dysfunction (10.3%). Roughly half of the patients (*n* = 44; 45.4%) presented no neurological symptoms. We attain a 14-month median imaging follow-up period and 16-month median clinical follow-up period. Chemotherapy (75.3%) and TKIs (68%) were administered before or after surgery and SRS. The EGFR-TKI agents used in this series included gefitinib (Iressa^®^), erlotinib (Tarceva^®^), afatinib (Giotrif^®^), and osimertinib (Tagrisso^®^). Among the 49 patients presenting EGFR mutations, 8 had taken Iressa, 31 had received Tarceva, and 15 had received Giotif. Tagrisso had been taken by only 5 patients, due presumably to a lack of insurance coverage for this medication. Many of the patients had previously taken more than two EGFR-TKIs. 

The most frequent tumor origin was pure adenocarcinoma (91.8%). Among the 89 patients with this tumor origin, 38 (36.9%) had wild-type EGFR (no mutation) and the other cases were involved with EGFR mutations. Note that 15 of the patients (15.5%) did not undergo an EGFR gene mutation test due to the late adoption of this technology by our institution. Most of the mutations were located at Exon 19 (18.4%), followed by L858R (14.6%) and L858R + T790M (3.9%). Other types of mutations, including Exon 21 deletions and mutations located at L816Q, T790M, and S7681, were observed in fewer than 1% of the cases. 

### 2.2. Surgical Resection

The patients in this study included only those for whom surgery was recommended as an alternative to radiation as the primary treatment. Prior to surgery, all patients underwent a physical examination, an MRI with Gadolinium, and a complete blood workup. Most of the patients underwent a body bone scan and contrast-enhanced CT or positive emission tomography scans of the chest, abdomen, and pelvis to determine the extent of extracranial malignant disease. After the pretreatment evaluation, all patients underwent a craniotomy with the goal of total metastases removal. All patients also underwent MRI to confirm that the surgical removal of the tumors was complete. 

### 2.3. SRS Procedures

SRS was performed using a Leksell Gamma Unit 4C or the Perfexion Stereotactic Radiosurgery device (Elekta AB, Stockholm, Sweden). The techniques used in our institution are detailed in previous papers [4,5,6,7,8]. Briefly, patients underwent thin-slice stereotactic MRI following stereotactic Leksell frame placement under monitored anesthesia. Dose planning was performed using Gamma Plan software. We recorded all SRS treatment parameters, including treatment volume, margin, and maximum dose. We calculated the tumor volume and maximum tumor diameter during the stereotactic MRI performed on the day of the SRS procedure.

### 2.4. Radiation Dose Scheme

The dose scheme was established in accordance with recommendations from the RTOG and Japanese study group. For patients undergoing adjuvant SRS, dose planning was conducted in accordance with RTOG 95–08 [9], wherein the SRS boost dose was adjusted to the size of the lesion, as follows: <20 mm (24 Gy), 20–30 mm (18 Gy), and 30–40 mm (15 Gy). In accordance with a previous prospective randomized multicenter phase III study, [10] SRS dose was adjusted to the size of the lesion, as follows: <20 mm (22–25 Gy) and >20 mm (18–20 Gy). In accordance with RTOG 90–05 [11] for previously irradiated brain metastases, the dose was adjusted to the maximum diameter of the lesion, as follows: ≤20 mm (18 Gy), 21–30 mm (15 Gy), and 31–40 mm (12 Gy).

### 2.5. Outcomes and Follow-Up

We performed clinical and neuroimaging follow-ups at 3-month intervals for the detection of recurrent brain metastases. Clinical evaluations include neurological examinations and a review of symptoms related to chemotherapy/target therapy. Neuroimaging studies included a whole brain contrast-enhanced thin-slice MRI. All neuroimaging studies were reviewed by two well-trained neuroradiologists (HM Wu and CJ Lin) independently. Tumor volumes were derived from the sum of the areas contoured in each slice multiplied by the slice thickness. According to the trapezoidal rule formula, multiplying contours of the post-surgical cavity by the slice thickness (mostly 3 mm thickness in follow-up series) should limit the error in calculated volume to 10% or less, as long as accurate delineation is achieved in at least five slices. Tumor response was assessed by comparing a follow-up MRI to the MRI obtained at the time of SRS. Tumor responses were categorized as follows: regression (≥10% decrease in tumor volume), stable (<10% increase or decrease in tumor volume), or progression (≥10% increase in tumor volume). Tumor control was defined as stable tumor response and tumor regression [12].

### 2.6. Statistical Analysis

Descriptive statistics for continuous variables were reported as medians or means, and categorical variables were reported as frequencies or percentages. Categorical variables were compared using the Chi-square or Fisher’s exact tests, as appropriate. Continuous variables were compared using the independent student’s t-test with or without equal variation, as appropriate. Time-dependent analysis of progression-free survival (i.e., tumor control) and overall survival (OS) were assessed using Kaplan–Meier and actuarial methods. Progression-free survival curves were compared using the log-rank test. Statistical significance was defined at *p* < 0.05, and all tests were two-tailed. All statistical analysis was performed using SPSS (version 20.0, IBM Corporation, Armonk, NY, USA). 

## 3. Results

### 3.1. Tumor Response and Overall Survival Following GKRS

Among patients with NSCLC-BM, the median imaging follow-up period after radiation treatment was 14 months. During the first 6 months of follow-up, 87.4% of the NSCLC-BM patients remained in a stable or regressive state, while 12.6% were in progression. Tumor control rates were as reported as 12 months (75.0%), 18 months (68.1%), 24 months (61.2%), 30 months (61.2%), and 36 months (57.1%), Figure 1e.

Among patients with NSCLC-BM, the median clinical follow-up after radiation was 16 months, and the median survival was 25 months. Actuarial overall survival rates were 89.6% (12 months), 79.6% (18 months), 73.9% (24 months), 70.6% (30 months), and 61.2% (36 months), Figure 1a.

### 3.2. Prognostic Factors Associated with Tumor Control

Multivariate analysis based on the Cox regression model revealed a correlation between post-craniotomy radiation (*p* = 0.027) and improved tumor control. As shown in Figure 1, the tumor control rates in the post-craniotomy radiation versus non-post-craniotomy radiation groups were 83.1% vs. 57.7% (12 months), 76.3% vs. 50.5% (18 months), 66.1% vs. 50.5% (24 months), 66.1% vs. 50.5% (30 months), and 60.1% vs. 50.5% (36 months), Figure 1f. 

We observed no difference between the EGFR wild-type group and the EGFR mutation group in terms of tumor control rates, Figure 1h. The 1-year tumor control rates were 81.6% (EGFR wild-type group) and 92.8% (EGFR mutation group).

Tumor control was not correlated with TKI use, chemotherapy use, tumor margin, or the presence of post-craniotomy residual tumor, Table 2.

### 3.3. Prognostic Factors Associated with Overall Survival

Multivariate analysis based on the Cox regression model for OS revealed a correlation between success in controlling the original tumor and longer survival durations. Post-craniotomy radiation was not a determining factor (*p* = 0.585). As shown in Figure 1b, the OS rates among cases with original tumor control were 95.9% (12 months), 81.4% (18 months), 81.4% (24 months), 78.6% (30 months), and 78.6% (36 months). The OS of these patients was better than that of patients without original tumor control.

The incidence of extracranial metastasis and the post-craniotomy residual tumor was not correlated with TKI use, chemotherapy use, or GPA score Table 3.

### 3.4. Demonstration Case

A 68-year-old male suffered from progressive left limb weakness and unsteady gait. The brain MRI in Figure 2a revealed a 41 mm mass in the medial portion of the right premotor and primary motor areas. Due to a personal history of cigarette exposure and accompanying respiratory symptoms, lung cancer was highly suspected. The pathology revealed in the CT-guided biopsy revealed EGFR wild-type lung adenocarcinoma (stage cT2bN0M1). A brain metastasis removed from the right frontal region revealed metastatic adenocarcinoma. A post-craniotomy MRI revealed no evidence of residual tumor in the surgical cavity, Figure 2b. One month after the craniotomy, the patient underwent GKS (Figure 2c) involving a marginal dose of 15 Gy and a maximum dose of 30 Gy to a surgical cavity with a volume of 21.5 mL. In MRI imaging follow-up, the surgical cavity showed tumor regression, Figure 2d,e. MRI imaging during a 9-month follow-up revealed a new lesion in the right middle frontal gyrus and a 12-month imaging follow-up revealed newly developed brain metastases, despite the fact that the post-GKS surgical cavity remained in a stable state, Figure 2f,g. The patient subsequently underwent a second GKS procedure (10 months post-craniotomy) for the removal of the newly developed brain metastases without salvage GKS to the surgical cavity (Figure 2h). MRI follow-up revealed that the patient remained in a stable condition until the end of the study period (15 months). Figure 2i,j. This case demonstrates that despite using GKS to irradiate the surgical cavity without any signs of residual tumor, it was not possible to prevent distant intracranial recurrence leading to secondary GKS or even WBRT.

## 4. Discussion

This retrospective study examined the use of SRS or WBRT to eradicate tumor tissue in post-craniotomy surgical cavities following the surgical resection of one or two metastases. Post-surgical radiation therapy was shown to improve outcomes in terms of local tumor control, compared to patients who did not undergo the procedure and patients for whom radiation therapy was delayed. Post-surgical radiation therapy was the only prognostic factor associated with improved local tumor control, regardless of the histologic type of non-small cell lung cancer, EGFR mutation state, resected tumor volume, or chemotherapy usage. 

We observed a positive correlation between original tumor control and the length of overall survival; however, we did not observe a significant correlation between post-craniotomy irradiation of the surgical cavity and overall survival. A previous randomized controlled phase 3 trial by Mahajan et al. [13] revealed similar outcomes, wherein the median overall survival was 18 months in the observation group and 17 months in the SRS group (*p* = 0.24). Qin et al. also reported that SRS provided no survival benefits among patients with resectable brain metastases [14].

Previous studies have reported that irradiating the surgical cavity can improve local tumor control [15,16]. It also appears that WBRT and SRS produce similar results in terms of local tumor control and overall survival. In a previous multicenter, randomized, phase 3 trial (Postoperative Stereotactic Radiosurgery Compared with Whole Brain Radiotherapy for Resected Metastatic Brain Disease; NCCTG N107C/CEC·3) [17], the median overall survival rates were 12.2 months (SRS) and 11.6 months (WBRT). In the same trial, the local control rates were 80.4% (SRS) and 87.1% (WBRT). Nonetheless, a number of studies have reported that SRS is more effective in preserving neurocognitive function and quality of life [18,19,20].

In an animal model by Das et al. [21], the clonogenic survival of mutant EGFR NSCLCs in response to ionizing radiation was 500- to 1000-fold higher than that of wild-type (WT) EGFR NSCLCs, which indicates that the EGFR mutation group was more sensitive to radiation than was the wild-type EGFR group. In a study by Gow et al. [22], the EGFR mutation group was significantly more responsive to WBRT than the EGFR wild-type group (54% vs. 24%, *p* = 0.045). Yang et al. [23] reported similar results in which the response to SRS and progression-free survival (*p* = 0.048) were higher in the EGFR mutation group than in the wild-type group.

In a previous study [24], we determined that the 1-year tumor control rate after SRS was higher in the EGFR mutation group (90.5%) than in the EGFR wild-type group (79.4%), and EGFR mutation status was correlated with overall survival. We also found that EGFR mutation was a prognostic factor for tumor control and that prior craniotomy was correlated to overall survival. Thus, our objective in the current study was to determine whether EGFR mutation was prognostic for tumor control or survival within the subgroup of patients who had undergone craniotomy. Note that we did not observe a correlation between local tumor control and EGFR mutation in this study. Note also that Wang et al. and Shin et al. reported no correlation between EGFR mutation status or local tumor control after SRS [25,26]. Clearly, further research will be required to determine whether EGFR mutation plays a role in the outcomes of radiation therapy.

Recent medications developed for EGFR mutations have improved overall survival and tumor control, and the fact that EGFR-TKI is able to penetrate the blood–brain barrier [27,28,29] could have a profound effect on the formulation of treatment strategies. We previously found that TKI usage was not correlated with overall survival or tumor control. However, other researchers have reported that among patients with the EGFR mutation, TKI-naive patients had longer overall survival and lower distant intracranial failure than their counterparts who had already started TKI therapy [30,31]. In the current study, we were not concerned with the timing of TKI treatment initiation; therefore, it is possible that our study cohort included EGFR mutation patients who had begun TKI therapy prior to radiation as well as those who had not. Further research will be required to determine whether TKI usage is prognostic to overall survival and tumor control rate. Administering EGFR-TKIs to patients scheduled for surgical resection would presumably reduce the differences in overall survival and local tumor control between patients with EGFR mutations and those without. Thus, further research will be required to elucidate the effects of TKI usage on local tumor control after SRS.

Many previous studies have reported that the inclusion of follow-up radiation after resection surgery can enhance tumor control rates. Note, however, that the interval between a craniotomy and SRS differed in different institutions, Table 4. Robbins et al. [32] reported a 1-year local control rate of 81.4% in cases where the interval between craniotomy and SRS did not exceed 8 weeks. Lorio-mortin et al. [33] reported a 1-year local control rate of 73% in cases where the interval between craniotomy and SRS did not exceed 3 weeks. In most previous studies, the interval between craniotomy and SRS was 2 to 8 weeks, and the corresponding 1-year local tumor control rate was 73% to 87%. [32,33,34,35,36] Strauss et al. [36] reported that a shorter interval between surgery and SRS was associated with better local control (*p* = 0.02). In the current study, the interval between craniotomy and radiation therapy was less than 3 months and the 1-year local tumor control rate was 83.1%, echoing the results in previous studies. Taken together, it appears that applying SRS to surgical cavities within 3 months is an effective approach to enhancing the local control rate.

### Study Limitations

This study had a number of limitations that should be considered in interpreting our findings. First, our analysis was subject to the shortcomings inherent to a retrospective study design. The selection of EGFR-TKI depended on the preference of the physician, and as a non-controlled study, the radiation and TKI doses were highly variable. It is also likely that in a few cases, the gene presentation of NSCLC-BMs might have changed after receiving target therapy. Lastly, the results are not necessarily generalizable to patients with other forms of brain metastases, such as tumors that do not respond well to radiation (e.g., renal cell carcinoma). Large cohort studies or randomized trials will still be required to prove the efficacy and safety of post-surgical cavity SRS.

## 5. Conclusions

Our results indicate that post-craniotomy SRS within three months of surgical resection is associated with improved local tumor control. In this study, only original tumor control was predictive for overall survival. EGFR mutation status was not significant for predicting overall survival or tumor control.

## Figures and Tables

**Figure 1 life-13-00236-f001:**
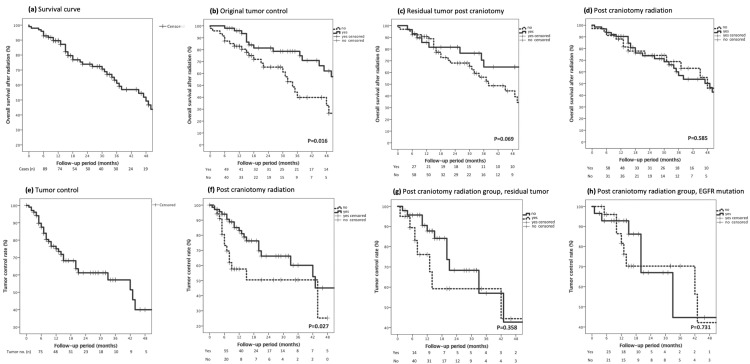
Kaplan–Meier analysis of overall survival (OS) rates among patients with non-small cell lung cancer brain metastasis (NSCLC-BM) who underwent radiation (Stereotactic Radiosurgery (SRS) or Whole Brain Radiotherapy (WBRT)): (**a**) OS; (**b**) outcomes of original tumor control; (**c**) outcomes of residual tumor post-craniotomy; and (**d**) outcomes of post-craniotomy radiation. Kaplan–Meier analysis of tumor control rates among patients with NSCLC-BM who underwent radiation (SRS or WBRT): (**e**) tumor control rate; (**f**) outcomes of post-craniotomy radiation; (**g**) outcomes of residual tumor post-craniotomy in post-craniotomy radiation group; and (**h**) influence of EGFR mutation in post-craniotomy radiation group.

**Figure 2 life-13-00236-f002:**
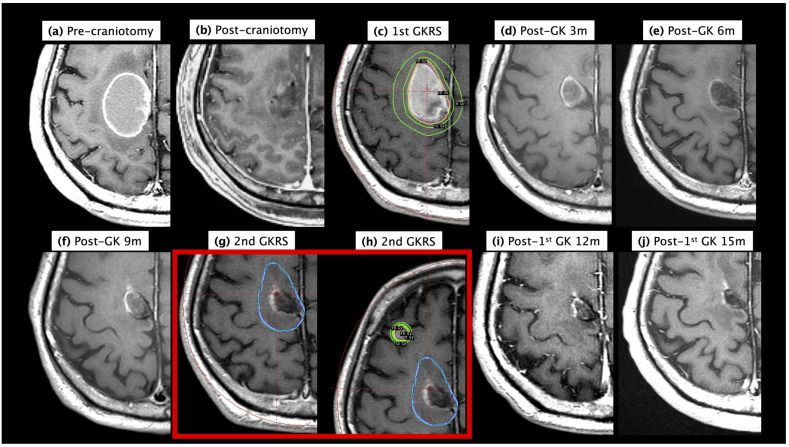
Case study of tumor local control based on 15-month image follow-up after post-craniotomy SRS: (**a**) pre-craniotomy contrast MRI; (**b**) post-craniotomy contrast MRI; (**c**) 1st Gamma Knife (GK) plan for post-craniotomy surgical cavity at one month after craniotomy; (**d**) tumor regression during 3-month follow-up after GKS; (**e**) tumor under stable control at 6 months after GKS; (**f**) tumor control at 9-month follow-up; (**g**) 2nd GK plan for newly developed brain metastasis with previously GK-treated post-craniotomy tumor under control at 10 months after craniotomy; (**h**) 2nd GK plan for newly developed brain metastases; (**i**) tumor control at 12 months after 1st GKS; and (**j**) tumor control at 15 months after 1st GKS.

**Table 1 life-13-00236-t001:** Clinical characteristics of 103 brain metastatic surgical cavities in 97 patients with non-small cell lung cancer.

Characteristics	Value	Percentage or Range
**Per patient (*n* = 97)**		
**Sex (Male:Female)**	49:48	
**Age at time of SRS (median, min, max)**	59.20	29.2–80.2
**Median max tumor vol. (mL)**		
**Multiple or solitary brain metastasis at SRS**		
**Solitary**	29	29.9%
**2–3**	32	33.0%
**4–10**	30	30.9%
**>10**	6	6.2%
**Numbers of surgical cavities**		
**1**	91	93.8%
**>1**	6	6.2%
**Neurological status**		
**Long tract sign**	41	42.3%
**Cerebellar sign**	0	0%
**Cranial nerve sign**	10	10.3%
**High cortical dysfunction**	10	10.3%
**Asymptomatic**	44	45.4%
**KPS score (median)**	90	50–100
**GPA score**		
**GPA 0–1**	2	2.1%
**GPA 1.5–2**	26	26.8%
**GPA 2.5–3**	39	40.2%
**GPA 3.5–4**	30	30.9%
**Median image follow-up (months)**	14	0–239
**Median clinical follow-up (months)**	16	0–241
**Median survival (months)**	25	0–241
**Per tumor (*n* = 103)**		
**Non-small cell lung cancer pathology**		
**Adenocarcinoma**	89	86.4%
**Squamous cell carcinoma**	5	4.9%
**Neuroendocrine tumor**	1	1.0%
**Poorly differentiated carcinoma**	2	1.9%
**Pleomorphic carcinoma**	1	1.0%
**AdenoCA + large cell neuroendocrine CA.**	1	1.0%
**AdenoCA + squamous cell CA.**	2	1.9%
**Inconclusive**	2	1.9%
**EGFR mutation type**		
**EGFR wild-type**	38	36.9%
**EGFR mutation**	49	47.6%
**Exon 19 deletion**	19	18.4%
**Exon 21 deletion**	1	1.0%
**L858R point mutation**	15	14.6%
**T790M point mutation**	1	1.0%
**S7681I point mutation**	1	1.0%
**L861Q point mutation**	1	1.0%
**Combined mutations**		
**Exon 20 Q878Q + L858R mutation**	1	1.0%
**L858R + T790M**	4	3.9%
**Wild-type + Exon 19 deletion**	2	2.0%
**G719 + S7681I**	1	1.0%
**Exon 19 deletion + T790M**	3	2.9%
**EGFR wild-type + ALK + ROS1**	1	1.0%
**ALK mutation**	2	2.0%
**Inconclusive**	15	13.4%
**Original tumor control**	52	53.6%
**Other metastases**	39	40.2%
**Chemotherapy use**	73	75.3%
**Target therapy use**	70	68%
**Prior WBRT**	34	35.1%
**Interval of lung ca. diagnosis to brain meta (months)**	0	0–115
**Craniotomy (*n* = 103)**		
**Gross total resection**	69	67.0%
**Subtotal resection**	30	29.1%
**Unknown**	4	3.9%
**Interval of craniotomy to SRS**		
**<3 months**	68	66.0%
**>3 months**	35	34.0%
**Location of tumor (at SRS)**		
**op bed**	58	56.3%
**Other sites**	45	43.7%
**Tumor volume (median, min, max)**	3.31	1.41–6.79
**SRS protocol**		
**Surgical cavity volume (TV1, mL)**	7.75	1.18–42.38
**Margin dose (TP, Gy)**	17	12–20
**Maximum dose (TC, Gy)**	32	21.4–40

**Table 2 life-13-00236-t002:** Prognostic factors associated with tumor control.

Factors	Tumor Control					
	Univariate			Multivariate		
	*p* Value	HR	95% Cl	*p* Value	HR	95% Cl
Age at time of GK (yrs)	0.189	0.978	0.946–1.011			
Sex (male vs. female)	0.885	1.054	0.520–2.136			
EGFR mutation (yes vs. no)	0.855	0.930	0.430–2.014			
Original tumor control (yes vs. no)	0.551	1.245	0.605–2.562			
Other metastasis (yes vs. no)	0.135	0.544	0.244–1.210	0.143	0.550	0.247–1.225
Chemotherapy (yes vs. no)	0.394	1.440	0.622–3.335			
EGFR-TKI use (yes vs. no)	0.175	1.628	0.805–3.292			
Interval of lung CA to brain metastases	0.571	0.996	0.980–1.011			
Number of lesions	0.477	0.958	0.851–1.078			
Post-craniotomy residual tumor (yes vs. no)	0.914	0.959	0.448–2.054			
Post-craniotomy radiation (yes vs. no)	**0.033**	2.137	1.065–4.286	**0.035**	2.116	1.055–4.243
TP margin dose	0.449	0.915	0.728–1.151			
TC maximum dose	0.195	0.935	0.845–1.035			
Tumor volume (mL)	0.199	0.962	0.906–1.021			

Boldface type is used to indicate statistical significance at *p* < 0.05. We included regressive and stable brainstem tumors as tumor control. Cox regression analysis was used to analyze factors that were potentially associated with local tumor control.

**Table 3 life-13-00236-t003:** Prognostic factors associated with overall survival.

Factors	Overall Survival					
	Univariate			Multivariate		
	*p* Value	HR	95% Cl	*p* Value	HR	95% Cl
Age (yrs)	0.589	0.992	0.962–1.022			
Sex (male vs. female)	0.076	0.560	0.295–1.063	0.051	0.515	0.265–1.002
GPA score (>=4, <4)	0.145	0.521	0.217–1.251	0.238	0.582	0.237–1.430
EGFR mutation (yes vs. no)	0.667	1.179	0.557–2.498			
Adenocarcinoma (yes vs. no)	0.180	1.752	0.772–3.977			
Original tumor control (yes vs. no)	**0.019**	2.112	1.129–3.951	**0.014**	2.273	1.181–4.374
Other metastasis (yes vs. no)	0.583	0.828	0.422–1.624			
Chemotherapy use (yes vs. no)	0.394	1.429	0.629–3.245			
EGFR-TKI use (yes vs. no)	0.779	1.097	0.574–2.098			
Number of lesions	0.279	1.046	0.964–1.134			
Post-craniotomy residual tumor (yes vs. no)	0.077	0.508	0.240–1.075	0.076	0.502	0.234–1.076
Post-craniotomy radiation (yes vs. no)	0.588	1.194	0.629–2.267			

Boldface type is used to indicate statistical significance at *p* < 0.05. We included regressive and stable brainstem tumors as tumor control. Cox regression analysis was used to analyze factors that were potentially associated with local tumor control.

**Table 4 life-13-00236-t004:** The previous literature related to post-operative surgical cavity Gamma Knife (GK) Radiosurgery.

Author	Pts	Cavities	Margin	Technology	Dose (Median Margin)	Median Overall Survival (OS)	Prognostic Factors for OS	1 Year Local Control (LC)	Prognostic Factors for LC
**2008 Soltys** [37]	72	76		cyberknife	18.6 Gy	15.1	RPA class1, extracranial metastasis	79%	increase conformality index
**2008 Mathieu** [38]	40	40	1 mm	GK	16 Gy	13	X	73% (13 mo)	X
**2009 Karlovits** [39]	52	52		Linac	15 Gy	15	No extracranial disease, solitary intracranial metastasis	LC: 92.3% (at median follow-up 14 months; no local control at 1 year)	X
**2009 Do** [40]	30	33	1–3 mm	linac	16 Gy	12	X	82% for local recurrence-free survival,	X
**2010 Hwang** [41]	25	25		GK	15–20 Gy	15	Distant recurrence	100	X
**2011 Jensen** [42]	106	112	1 mm	GK	17 Gy	10.9	X	80.3%	pre-operative tumor > 3 cm
**2011 Rwigema** [43]	77	89	1 mm	cyberknife	18 Gy	14.5	X	76.1%	X
**2012 Prabhu** [44]	62	64	0–2 mm	Linac	18 Gy	13	X	78%	small PTV, marginal dose < 18 Gy
**2012 Robbins** [32]	85	85	2–3 mm	LINAC	16 Gy	12.1	Longer cancer to brain met time, solidary tumor	81.4%	target volume > 15 cm^3^, marginal dose of < 16 Gy
**2013 Luther** [45]	120	120	2–3 mm	GK	16 Gy	NA	X	87%	PTV, cavity diameter, margin dose > 16 Gy
**2014 Lorio Mortin** [33]	110	113	1 mm	GK	18 Gy	63%	X	73%	short surgery-to-SRS interval (<3 week), greater max radiation dose
**2014 Brennan** [34]	49	50	2 mm	LINAC	18 Gy	12	X	78%	NSCLC-histology, tumor maximal diameter < 3 cm, deep parenchymal tumor
**2014 Ojerholm** [46]	91	96	0 mm	GK	16 Gy	22.3	Active extracranial disease	81%	pre-operative metastasis < 3 cm, no residual/recurrent tumor
**2015 Abel** [35]	85	85	0–2 mm	GK	17.3 Gy	20	Gross total resection	87%	X
**2015 Strauss** [36]	100	102	No margin	LINAC	20 Gy	18.9	Active systemic disease, RPA class, KPS, multiple brain lesions, volume of brain metastases	84%	shorter interval between surgery and SRS
**2016 Rava** [47]	85	85	1–2 mm	GK	18 Gy	14.3	X	82%	tumor diameter < 3 cm, resection cavity volume < 14 mL
**2017 Mahajan** [13]	64	64	1 mm	GK	16 Gy	17	Stable disease	72%	Metastatic size
**2017 Brown** [17]	98	98	2 mm	GK	12–18 Gy	12.2	X	60.5%	SRS: better preservation of neurocognition, QoL

## Data Availability

Data are available from the Taipei Veteran General Hospital Gamma Knife Database. Due to legal restrictions imposed by the Taipei Veterans General Hospital in relation to the “Personal Information Protection Act”, data cannot be made publicly available.

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
