# Peer review of "Gamma Knife Radiosurgery Irradiation of Surgical Cavity of Brain Metastases: Factor Analysis and Gene Mutations"

_life, 2023, doi:10.3390/life13010236_

Round 1

Reviewer 1 Report

Gamma Knife radiosurgery irradiation of surgical cavity of 2 brain metastases: Factor analysis and gene mutations

General comments

The authors retrospectively review the effect of post-surgery radiation for brain metastases in 97 patients and try to discern effects of certain gene mutations, chemotherapy and TKI use on outcome.  The only correlations found are between tumor control and post-surgery radiation, and control of the original tumor and overall survival.  There was no correlation between post surgery radiation and overall survival, which is consistent with other studies. 

A lot of variables are being considered in this study.  It would be helpful to include a statement about how well this study was powered to discern differences, if any.

The manuscript is well written, but the conclusion appears incorrect.  There is no evidence in this data that post-craniotomy SRS was predictive of overall survival, as shown in figure 1d and Table 3.  This discrepancy needs to be resolved. 

Specific comments

L 29  replace ‘however’ with ‘hence’

Table 1  no information about tumor volume

L 113  Stereotractic

Section 2.4  Please state more clearly what the different dose schemes are used for; I presume the first is for the post-surgical cavity, the second for other metastases, the third (this one was clearly stated) for re-irradiation of metastases.

L 134 metastases

Figure 1  poor legends; titles or descriptions could also be better.  Example:  (c) Residual tumor post craniotomy: do you mean incomplete resection or is this referring to a regrown tumour?

Table 2

p Value of 0.134 does not need to be in bold type

Table 3

Last p Value in the list has a superscript 1, which suggests a footnote of some sort.  I don’t see a footnote labelled with ‘1’.

Table 4

Matheiu – incorrect spelling

L 321  SRS prior or post-surgery are very different.  If you are analyzing a mix, then this needs to be stated clearly much earlier.  More to the point, if you know when SRS took place relative to surgery, then this should be its own stratification.

Conclusion

“post-craniotomy 331 SRS was an independent predictive factor of overall survival”

This statement is not consistent with figure 1d and the information in Table 3 (last entry).

Your conclusion is therefore not correct.

Author Response

Response to Reviewer 1 Comments

Gamma Knife radiosurgery irradiation of surgical cavity of brain metastases: Factor analysis and gene mutations

General comments

The authors retrospectively review the effect of post-surgery radiation for brain metastases in 97 patients and try to discern effects of certain gene mutations, chemotherapy and TKI use on outcome.  The only correlations found are between tumor control and post-surgery radiation, and control of the original tumor and overall survival.  There was no correlation between post surgery radiation and overall survival, which is consistent with other studies.  A lot of variables are being considered in this study.  It would be helpful to include a statement about how well this study was powered to discern differences, if any. The manuscript is well written, but the conclusion appears incorrect.  There is no evidence in this data that post-craniotomy SRS was predictive of overall survival, as shown in figure 1d and Table 3.  This discrepancy needs to be resolved.  

Response: Thank you for pointing out the mistakenly wrote sentence, we have rearranged the composition to correlate to our findings. (page 12)

Specific comments

L 29  replace ‘however’ with ‘hence’

Response: Thank you for the notification, I have modified the writing. (page 1)

Table 1  no information about tumor volume

Response: Thank you for the notification, I have updated the information of tumor volumes in Table1. (Table 1 and page 4)

L 113  Stereotractic

Response: Thank you for the notification, I have modified the writing. (page 5)

Section 2.4  Please state more clearly what the different dose schemes are used for; I presume the first is for the post-surgical cavity, the second for other metastases, the third (this one was clearly stated) for re-irradiation of metastases.

Response: For regular surgical cavity SRS, we followed the RTOG 95-08 guideline, and a bit adjusted the dose considering tumor pathology, gene presentation, and target-/immune-therapy usage; for the other metastases without resection, we usually give them 18-25Gy according to the tumor size. For the repeated radiation for brain metastasis, we didn’t decrease the radiation when the tumor recurrence is outfield. On the other hands, we tapered the radiation dose when the tumor recurrence is infield (local recurrence) when we performed repeated SRS. (page 5)

L 134 metastases

Response: Thank you for the notification, I have modified the writing.

Figure 1  poor legends; titles or descriptions could also be better.  Example:  (c) Residual tumor post craniotomy: do you mean incomplete resection or is this referring to a regrown tumour?

Response: I revised the description to post craniotomy residual tumor for a better declaration of the incomplete resection during the craniotomy. (Figure 1)

Table 2

p Value of 0.134 does not need to be in bold type

Response: I have modified the mistakenly bold type. (page 7)

Table 3

Last p Value in the list has a superscript 1, which suggests a footnote of some sort.  I don’t see a footnote labelled with ‘1’.

 Response: Thank you for the notification, I have removed the mistakenly placed footnote. (Table 3)

Table 4

Matheiu – incorrect spelling

Response: Thank you for the notification, I have modified the spelling. (page 11)

L 321  SRS prior or post-surgery are very different.  If you are analyzing a mix, then this needs to be stated clearly much earlier.  More to the point, if you know when SRS took place relative to surgery, then this should be its own stratification.

Response: We have rechecked our database and confirmed that all patients under enrollment received SRS post-surgery, so we deleted the sentence. (page 12)

Conclusion

“post-craniotomy 331 SRS was an independent predictive factor of overall survival”

This statement is not consistent with figure 1d and the information in Table 3 (last entry).

Your conclusion is therefore not correct.

Response: Thank you for pointing out the mistakenly wrote sentence, we have rearranged the composition to correlate to our findings. (page 12)

Reviewer 2 Report

The manuscript entitled “Gamma Knife radiosurgery irradiation of surgical cavity of 2

brain metastases: Factor analysis and gene mutations” (MDPI life-2098742) by Y. Huang reports on a retrospective study aimed at comparing the effectiveness of post-surgical SRS to the cavity as adjuvant to surgery in brain metastases cases. Authors have evaluated the local tumor control and the overall survival as representative outputs.

The manuscript addresses a relevant issue, and it is clearly reported. I have only very minor -perhaps irrelevant- features to point out, which are listed below:

2.6. Statistical analysis:

1.     Are morphological imaging techniques capable of ensuring progression-free survival evaluation? (For instance, sentences like “… Post-craniotomy MRI revealed no evidence of residual tumor in the surgical cavity…” appear to be purely based on anatomical information) Why no functional imaging? Authors are invited to discuss/support their proposal

2.     Information, legends, axes values/names in Figure 1 are difficult to visualize.

2.4 Demonstration case:

3.     According to the text and the Figure 2, treatment planning has been performed on the pre-craniotomy patient anatomy instead of the post-craniotomy cavity. Is that correct? Moreover, follow-up Figures 2D and 2E show tumor volume regression. Maybe this reviewer has misunderstood the issue, but did the patient craniotomy involve tumor extraction/removal? The first sentence in the Discussion section claims “… This retrospective study examined the use of SRS or WBRT to eradicate tumor tissue in post-craniotomy surgical cavities following the surgical resection of one or two metastases….”, thus suggesting or directly indicating that tumors are removed by craniotomy surgery.

Author Response

Response to Reviewer 2 Comments

2.6. Statistical analysis:

  1. Are morphological imaging techniques capable of ensuring progression-free survival evaluation? (For instance, sentences like “… Post-craniotomy MRI revealed no evidence of residual tumor in the surgical cavity…” appear to be purely based on anatomical information) Why no functional imaging? Authors are invited to discuss/support their proposal

Response: Except the regular T1WI, T2WI, and T1WI with contrast. ,we also performed diffusion weight imaging (DWI) in our regular post craniotomy MRI checkups to assist interpretation of residual tumor or not, while the most accurate way of ensuring whether the residual tumor exists should be claimed by the surgeon under microscopic visions during the surgery. There are still limitations on post craniotomy MRI to identify the residual tumor, so we usually combined the surgeon’s note, T1/T2/T1+c/DWI images for residual tumor evaluation. In addition, functional imaging wasn’t performed due to lack of cost efficiency. We add this issue in our study limitation (page 8)

  1. Information, legends, axes values/names in Figure 1 are difficult to visualize.

Response: Thank you for pointing out this problem, we will try to sent high-resolution Figure1 to the Journal office email box and please help us to check.

2.4 Demonstration case:

  1. According to the text and the Figure 2, treatment planning has been performed on the pre-craniotomy patient anatomy instead of the post-craniotomy cavity. Is that correct? Moreover, follow-up Figures 2D and 2E show tumor volume regression. Maybe this reviewer has misunderstood the issue, but did the patient craniotomy involve tumor extraction/removal? The first sentence in the Discussion section claims “… This retrospective study examined the use of SRS or WBRT to eradicate tumor tissue in post-craniotomy surgical cavities following the surgical resectionof one or two metastases….”, thus suggesting or directly indicating that tumors are removed by craniotomy surgery.

Response: Treatment planning was performed on the post craniotomy MRI which was shot on the same day before the patient received SRS. Figure 2(b) was performed a few days right after the craniotomy, while the figure 2(c) was done a few weeks post craniotomy, therefore showing hyperdensity that was highly suspected caused by edema rather than tumor tissue. Figure 2(d) and 2(e ) both showed regression that is thought to be a desired result of SRS for tumor control. The claims of the discussion suggesting that tumors removed by craniotomy should still receive SRS to their surgical cavity for better tumor control, which is correlated to our demonstration case.

Round 2

Reviewer 1 Report

Gamma Knife radiosurgery irradiation of surgical cavity of 2 brain metastases: Factor analysis and gene mutations

Review 2

The manuscript has improved.  You spend quite a lot of time reviewing gene mutation information, which is appropriate given the title of the manuscript.  However, this does not get mentioned in the abstract or introduction, leading readers to believe this is a just another study showing surgical bed radiation is advisable for tumour control.  I strongly advice you include some information about your gene mutation work in the abstract and introduction.  Similarly, please include some conclusion about gene mutation effects (or non-effects) in the conclusion.

Line 86-87.  16 months (space between number and word); use of the word ‘reported’ is a bit peculiar, it’s as if you reported this somewhere else.  Maybe use the verb ‘attain’ instead.

Line 169.  I prefer ‘as follows’, even if you use that repeatedly.  See my comment about the use of the verb ‘report’ above.  If you keep ‘reported as’, then remove the ‘as’ before it.

Line 173.  You state you want to report survival rates, so I suggest you rewrite as:  Actuarial overall survival rates were: 92.8% (12 months), 89.6% (18 months), etc.  Same applies to

Lines 187-189, 192, 205-206, 266-267.  See comment for line 173 – why put the information you want to show in brackets?

Table 3: p < 0.05 means factor was significant, so why are values greater than this shown in bold?

Line 259.  metastases (plural)

Line 283.  You state an objective in the discussion section that belongs in the abstract and introduction.

Line 285.  patients

Line 332.  Funny use of ‘at last’.  Try ‘And last’ or ‘The results are also not….’

Author Response

The manuscript has improved.  You spend quite a lot of time reviewing gene mutation information, which is appropriate given the title of the manuscript.  However, this does not get mentioned in the abstract or introduction, leading readers to believe this is a just another study showing surgical bed radiation is advisable for tumour control.  I strongly advice you include some information about your gene mutation work in the abstract and introduction.  Similarly, please include some conclusion about gene mutation effects (or non-effects) in the conclusion.

 Response: Thank you for your kindly advise, I have included the information of gene mutations in the manuscript. (page1, 2, 13)

Line 86-87.  16 months (space between number and word); use of the word ‘reported’ is a bit peculiar, it’s as if you reported this somewhere else.  Maybe use the verb ‘attain’ instead.

Response: I have modified the the word. (Page 4)

Line 169.  I prefer ‘as follows’, even if you use that repeatedly.  See my comment about the use of the verb ‘report’ above.  If you keep ‘reported as’, then remove the ‘as’ before it.

Response: Thank you for the suggestion, I have modified the word. (Page 6)

Line 173.  You state you want to report survival rates, so I suggest you rewrite as:  Actuarial overall survival rates were: 92.8% (12 months), 89.6% (18 months), etc.  Same applies to

Lines 187-189, 192, 205-206, 266-267.  See comment for line 173 – why put the information you want to show in brackets?

 Response: Thank you for the suggestion, I have modified the manuscript about the report of survival rate and tumor control. (page 6,7) While the discussion was intended to highlight the different response to radiotherapy in EGFR mutation group versus EGFR wild group, with response rate written as an supplementary information. (page 10)

Table 3: p < 0.05 means factor was significant, so why are values greater than this shown in bold?

Response: Thank you for pointing out the mistakenly bold typed values, I have modified them. (page 8)

Line 259.  metastases (plural)

Response: I have modified the manuscript. (page 4)

Line 283.  You state an objective in the discussion section that belongs in the abstract and introduction.

Response: Thank you for your kindly advise, I have included the information of gene mutations in the manuscript. (page1, 2, 13)

Line 285.  Patients

Response: I have modified the manuscript. (page 10)

Line 332.  Funny use of ‘at last’.  Try ‘And last’ or ‘The results are also not….’

Response: Thank you for the suggestion, I have modified the word. (page 12)
